# Hatching Success Rather Than Temperature-Dependent Sex Determination as the Main Driver of Olive Ridley (*Lepidochelys olivacea*) Nesting Activity in the Pacific Coast of Central America

**DOI:** 10.3390/ani11113168

**Published:** 2021-11-05

**Authors:** Alejandra Morales Mérida, Aude Helier, Adriana A. Cortés-Gómez, Marc Girondot

**Affiliations:** 1Laboratoire Écologie, Systématique, Évolution, Université Paris Saclay, AgroParisTech, Centre National de la Recherche Scientifique, 91405 Orsay, France; alemoralesmerida@gmail.com (A.M.M.); aude.helier@hotmail.fr (A.H.); adriana.niobe@gmail.com (A.A.C.-G.); 2Doctorado en Ciencias Naturales para el Desarrollo (DOCINADE), Instituto Tecnológico de Costa Rica (TEC), Universidad Nacional de Costa Rica (UNA), Universidad Estatal a Distancia (UNED), Heredia 40101, Costa Rica; 3Facultad de Ciencias Químicas y Farmacia, Universidad de San Carlos de Guatemala (USAC), Guatemala 01012, Guatemala

**Keywords:** temperature-dependent sex determination, hatching success, albedo, *Lepidochelys olivacea*, olive ridley, sea turtle

## Abstract

**Simple Summary:**

In marine turtles, sex is determined during a precise period during incubation: males are produced at lower temperatures and females at higher temperatures, a phenomenon called temperature-dependent sex determination. Most predictions about the long-term persistence of sea turtle populations in the face of climate change have focused on the effect of incubation temperature on sex ratios. In Central America, the alternations in dark sand beaches (hotter sand) and light sand beaches (cooler sand) are observed. Due to the higher production of females at high temperatures and the natal homing phenomenon in marine turtles, the largest proportion of nests on dark sand beaches was expected. However, the inverse was observed. We hypothesize that high beach temperatures, being seen in darker sand, increased female-biased primary sex ratios but reduced the output of female hatchlings due to embryo thermal lethality at high temperature. Our study reveals that when we think about sea turtle population dynamics, we should consider a variety of factors and not only sex ratio.

**Abstract:**

In marine turtles, sex is determined during a precise period during incubation: males are produced at lower temperatures and females at higher temperatures, a phenomenon called temperature-dependent sex determination. Nest temperature depends on many factors, including solar radiation. Albedo is the measure of the proportion of reflected solar radiation, and in terms of sand color, black sand absorbs the most energy, while white sand reflects more solar radiation. Based on this observation, darker sand beaches with higher temperatures should produce more females. As marine turtles show a high degree of philopatry, including natal homing, dark beaches should also produce more female hatchlings that return to nest when mature. When sand color is heterogeneous in a region, we hypothesize that darker beaches would have the most nests. Nevertheless, the high incubation temperature on beaches with a low albedo may result in low hatching success. Using Google Earth images and the SWOT database of nesting olive ridleys (*Lepidochelys olivacea*) in the Pacific coast of Mexico and Central America, we modeled sand color and nesting activity to test the hypothesis that darker beaches host larger concentrations of females because of feminization on darker beaches and female philopatry. We found the opposite result: the lower hatching success at beaches with a lower albedo could be the main driver of nesting activity heterogeneity for olive ridleys in Central America.

## 1. Introduction

Marine turtles are present on many tropical sandy beaches during their nesting periods. However, not all beaches in intertropical regions receive the same number of marine turtle nests; some beaches host high densities of nests, whereas others, sometimes located in the same region, may have very few nests [1]. The origin of this difference in nesting at nearby beaches may relate to several factors: (i) accessibility to the beach from the sea [2], (ii) social facilitation for finding a nesting beach [3,4], and (iii) female philopatry with past heterogeneous nesting activity [5]. Philopatry is the tendency of an organism to stay in or habitually return to a particular area [6]. Marine turtles show natal philopatry [7] and nesting philopatry both among available nesting beaches and within the same nesting beach [2]. Thus, the spatial heterogeneity of nesting density could result from differences in female production at beaches. If more juvenile females are produced at one beach, then due to natal homing, this beach would be expected to receive more female adults in the next generation, and this phenomenon will increase beach heterogeneity from year to year.

Two opposite phenomena relating to nest temperature can drive female production at beaches [8]. First, in marine turtles, the sexual phenotype of embryos is determined by incubation during the middle third of development (middle third of incubation at constant temperature): males are produced at lower temperatures and females at higher temperatures. This phenomenon is known as temperature-dependent sex determination. Second, embryo development can be hampered if incubation temperatures are too high or too low. The temperature range for development is between 25 and 35 °C for marine turtles [9]. In turn, nest temperature depends on many factors, such as the depth of the nest, vegetation cover [10], the sea temperature [11], and the soil-absorbed incident solar radiation [12]. Albedo is the measure of the diffuse reflection of solar radiation out of the total incident solar radiation and is measured on a scale from zero, corresponding to a black body that absorbs all incident radiation, to one, corresponding to a body that reflects all incident radiation. Beach sand comprises different materials of diverse origins. Consequently, sand can have a range of different colors, from white sand (of coral origin, for example) to black sand (of volcanic origin). Soil color can be used to predict albedo [13,14]. The effect of sand color on nest temperature was demonstrated: nests deposited in black sand are warmer than those in white sand [12]. As a result, more females should be produced on black-sand beaches that have higher temperatures if the nests are not too hot to produce hatchlings. When this observation is linked to female philopatry, we should expect higher nesting activity on beaches with dark sand.

Central America is a relatively recent geological formation (<3.5 million years) with many active volcanos [15]. This history is visible on its beaches: dark sand results from the erosion of recent volcanic eruptions, whereas lighter sand is due to the aggregation of organic material from the sea. Four marine turtle species nest on the Pacific beaches of Central America every year (hawksbills (*Eretmochelys imbricata*), olive ridleys (*Lepidochelys olivacea*), leatherbacks (*Dermochelys coriacea*), and green turtles (*Chelonia mydas*)). *L. olivacea* employs two nesting strategies depending on aggregation density: (1) solitary nesting similar to other species, and (2) the group or mass-nesting (arribada) behavior in which several thousand females simultaneously nest on the same beach [16]. In Central America, beaches have been monitored for several decades, and density maps of olive ridley crawls, nests, and nesting females are available in both the scientific and gray literature.

Thus, the alternating darker and lighter sand in Pacific Central America and the presence of beaches with varying densities of marine turtle nests represent an ideal situation to test the hypothesis that female production combined with natal homing is the driver of the heterogeneity of nesting activity. The aim of this study was to test whether a correlation exists between sand albedo and the nesting activity and to discriminate between the hypotheses that population dynamics are linked to temperature-dependent sex determination or to the deleterious effects of high temperature on incubation success.

## 2. Materials and Methods

### 2.1. Datasets

Beach images were searched for using Google Earth Pro V 7.3.2.5776 by visually examining the whole Central American Pacific coastline. A beach was defined as a continuous stretch of sand visible in satellite photography available from Google Earth. The Pacific coasts of Mexico, Guatemala, El Salvador, Costa Rica, and Panama were examined to identify the presence of beaches, resulting in a total of 291 beaches (Figure 1). Only the most recent images were used. For each beach, the coordinates of both ends, the image date, and the standardized color of the sand in the middle of the beach (see below for description) were recorded. The length of the beaches was calculated using haversine distance between the coordinates of both ends. View altitude was always chosen to display the image of the entire beach on a 15″ monitor. Volcano longitude and latitude were retrieved from the Smithsonian Institution’s online database of Holocene Volcanos [17]. For marine turtle density, data on crawls, nests, and nesting females along the Central American Pacific coast were retrieved from the State of the World’s Sea Turtles database online (SWOT; March 2020 version) [18] along with any relevant literature using the database information.

Data from different studies, from both dark and light beaches, that incubated eggs at constant temperature were extracted from the literature and are available in the DatabaseTSD file, as part of the R package embryogrowth [19]. The database (2021-09-16 version) includes 1456 records for 59 species. Only data from eggs incubated in temperature-regulated chambers were used. Regional management units (RMUs) for olive ridley sea turtles of the East Pacific and Atlantic West were retrieved [20,21,22,23,24,25,26,27,28,29,30]. RMUs were inferred from the marine turtle biogeography, including nesting sites, population abundances and trends, population genetics, and satellite telemetry [31]. The following variables were retrieved from the database: incubation temperature, total number of incubated eggs, number of hatched eggs, number of sexed turtles, and number of males and females.

### 2.2. Beach Color

The beach images captured on Google Earth (1024 × 768 pixels) were processed using Photofiltre software (version 7.2.1 accessed on 30 November 2005, http://www.photofiltre.com, Antonio Da Cruz, Houilles, France). Color histograms for the red, green, and blue (RGB) components of the pixels for a portion of the image can be extracted with this software.

A square of 85 × 85 pixels (7285 pixels) located at the center of each beach was analyzed. This square was chosen in the middle of the transect from the sea to the vegetation line of the beach so as to only include sand. The modal value for the color of the pixels in each square was calculated. The use of the center was justified by the observation that it is generally the zone with highest density of nests [32]. The lightest and darkest zones of the image that included the entire beach were then selected to represent the color endpoints to standardize color variability across beaches. The Euclidean distances between the RGB modes of the lightest and darkest zones of the image and the RGB modes of the center of the beach were calculated. The two resulting values were standardized to obtain a final value between 0 and 1 (0 being the value for a white range and 1 for a black range). These values are inversely related to the albedo of the sand. Colors were individually standardized for each image; thus, this methodology corrects for the time of day and the cloud cover when each photograph was taken.

Two tests were performed to evaluate the accuracy of the color estimation from Google Earth pictures. First, we checked that the color of beach sand estimated from Google Earth pictures showed a spatial structure. Second, we tested whether dark beaches were located closer to volcanos than lighter beaches, as expected because basalt material from volcanic origin is darker than material of non-volcanic origin. Color estimation was cross-checked with our personal observations some of these beaches (AMM: Guatemala; MG: Mexico, Guatemala, and Costa Rica; AACG: Mexico), and with a survey of the literature.

The relationship between the estimated darkness of the beach and the closest volcano was estimated using the Mantel test, which is a statistical test of correlation between two matrices. It is based on a linear correlation and thus subject to the same assumptions as the Pearson correlation. Because of this limitation, permutation methods are used for significance testing when assumptions of independence are not met. This is the case for spatially distributed information that is linked by their process of formation. For example, basalt material of volcanic origin can be present on a beach, and thus, beach material and volcano presence are not independent. The Mantel test was performed using 9999 permutations with the function mantel of the R Community Ecology Package vegan 2–6 [33].

### 2.3. Standardizing Nesting Activity

Quantitative nesting information was available for 90 beaches in Central America on the Pacific coast for the years ranging from 1997 to 2014 (1620 year–beach combinations). The number of nests was available for 169 combinations. To obtain an index of nesting for each nesting beach, a model of temporal and spatial nesting patterns in the region was built to estimate the proportion of nests for a beach based on the relative frequency of nests at the different beaches and the total number of nests for each year. The aim of this model was to define an index of the nesting activity for each beach in that region when the years with data were not the same for all the beaches.

Considering the total number of nests T_i_ for year i in the entire region (SWOT database) where K beaches were monitored for Y years, three different models can be used to describe T_i_ according to year i:Constant number of nests: T_i_ = T; one parameter, T;Exponential model: T_i_ = T_0_ e^r.i^, where the two parameters T_0_ and r are the number of nests at time 0 and the growth rate, respectively;Year-specific number of nests: T_i_; Y parameters, T_1_ to T_Y_.

The distribution of nests across the different beaches is defined by the proportion p_j_ of T_i_ nests on j beach. It should be noted that the p_j_ are constrained to be constant over time; thus, the index of nesting on each beach is the same for any year. For a total of K beaches, a total number of K–1 parameters *p* is necessary due to the relation $\sum_{j=1}^{K}p_{j}=1$. Using one model, the expected number of nests for year i on beach j is thus E_i,j_ = T_i_ × p_j_. The time-constant constraint about p_j_ is made necessary by the scarce information that was available (169 beach–year data points), which prevented a more complex model to be fitted.

Let N_i,j_ be the observed number of nests. During the fit, the standard deviation was modeled as a linear estimate of the observed number of nests: S_i,j_ = a N_i,j_ + b with a and b > 0 (two parameters), and a Gaussian distribution model was used. For the final estimate, the expected number of nests E_i,j_ was only used when no observation was available; in other situations, the number of observed nests N_i,j_ was preferred.

The −ln likelihood of the observations within the model is simply the sum of the −ln likelihood for each observation N_i,j_ within the Gaussian model $N\left(E_{i,j},S_{i,j}\right)$. The best fitting model for each dataset was selected based on the maximum likelihood. Model selection was performed based on the minimum Akaike information criterion (AIC) [34]. AIC measures the quality of the fit, which is simultaneously penalized for the number of parameters in the model. It facilitates the selection of the best compromise between fit quality and over-parametrization from a set of models. When a set of models is compared, it is possible to estimate the relative probability that each model is the best among those tested using the Akaike weight [35]. Maximum likelihood fitting of parameters was made using the R package *phenology* that implements this model [36].

### 2.4. Relationship between Sand Color and Nesting Activity

A linear model was used to test for the relationship between the index of nesting activity for each beach and the beach sand color and beach length. We used the log_10_ proportion of nests in different beaches to normalize data, and then a Gaussian distribution was used.

### 2.5. Thermal Reaction Norm for Hatching Success and Sex Ratio

Due to albedo change, incubation temperatures on dark-sand beaches are supposed to be higher than on white-sand beaches. Incubation temperature influences both hatching success and sex ratio. The fitting of the sex ratio thermal reaction norm for the East Pacific RMU was published in Abreu-Grobois et al. [37]. The methodology is recalled here briefly. Data on the number of males and females produced for incubations at 17 constant temperatures were used. The relationship between constant incubation temperature and sex ratio was fitted using the logistic equation [38], and the credible interval was fitted using the Metropolis-Hastings algorithm with a Monte Carlo Markov chain in Bayesian analyses with uniform priors see [37] for more details.

Data on the number of hatchlings produced at constant incubation temperatures are available for the Pacific East RMU (20 constant incubation temperatures from Costa Rica, Panama, and Mexico) and Atlantic West RMU (13 constant incubation temperatures from Brazil). These data were fitted using the scaled product of two logistic equations to model the observation that hatching success (*HS*) according to constant incubation temperature (*t*) is null at low and high temperatures.
(1)$$HS=MaxHS\times \frac{1}{1+e^{4\;\left(P_{low}-t\right)/S_{low}}}\times \frac{1}{1+e^{4\;\left(P_{low}+\Delta P-t\right)/S_{high}}},$$

*P_low_* and *S_low_* refer to the transition from 0 to *MaxHS* at lower temperatures, whereas *P_low_* + Δ*P* (with Δ*P* > 0) and *S_high_* refer to the transition from *MaxHS* to 0 at higher temperatures. In these equations, *P* is the temperature at which hatching success is 0.5, and *S* is the slope at *P*. The fitting was made using binomial distribution and maximum likelihood. The credible interval was fitted using the Metropolis-Hastings algorithm with a Monte Carlo Markov chain in Bayesian analyses with uniform priors $P_{low}\sim U\left(20;40\right)$, $S_{low}\sim U\left(0;5\right)$, $MaxHS\sim U\left(0;1\right)$, $\Delta P\sim U\left(0;10\right)$, and $S_{high}\sim U\left(-5;0\right)$; see [25] for more details of the statistical methodology. Maximum likelihood and Bayesian estimates were made using the R package embryogrowth that implements these models [19].

## 3. Results

### 3.1. Beach Albedo from Satellite Images

A Mantel test using matrices of distances and albedo differences among beaches showed a significant spatial organization of beach albedo (Mantel test, *p* = 0.02), indicating that two nearby beaches are more similar in albedo than expected from a random distribution.

The relationship between beach color and distance to the closest Holocene volcano is very strong (ΔAIC = 25.50, Akaike weight > 0.9999): sand albedo increases with the proximity of the nearest volcano.

### 3.2. Temporal Olive Ridley Nest Abundance in Pacific Central America

Different temporal models were tested. The Year-Specific (YS) model is the selected model (Table 1) with an Akaike weight of 0.94 indicating a strong support. The temporal and spatial results are shown in Figure 2, and the estimated proportion of nests (log_10_ scale) for the 90 nesting beaches with information is shown in Figure 1.

### 3.3. Relationship between Beach Albedo and Olive Ridley Nest Number and Density

We only used olive ridley nest counts from the SWOT database, as this species had the larger amount of data, both temporally and spatially. We used the log_10_ proportion of nests on different beaches to normalize data. The relationship between the log_10_ proportion of nests, number and density of nests per kilometer, and the sand darkness index was negative (Figure 3A,B); darker beaches tended to have less nesting activity than lighter beaches.

### 3.4. Thermal Reaction Norm for Sex Ratio and Hatching Success

The pivotal temperature (theoretical temperature that produces both sexes in equal proportion) for East Pacific olive ridleys is 30.24 °C (95% credible interval 30.04–30.50 °C), while the transitional range of temperatures 5% (TRT 5%, temperature range producing 5% to 95% of both sexes) is 3.84 °C (95% CI 3.08–4.72 °C) (Figure 4A). The lower and upper limits of TRT 5% are, respectively, 28.33 °C (95% CI 27.80–28.76 °C) and 32.16 °C (95% CI 31.71–32.68 °C).

The fitted thermal reaction norm for hatching success is shown in Figure 4B. It shows two abrupt declines below P_low_ = 24.83 °C (95% CI 23.19–24.98 °C) and above P_low_ + ΔP = 33.57 °C (95% CI 33.08–34.28 °C).

## 4. Discussion

Google Earth images have already been used in science for numerous applications [39]. We expanded this use to the study of beach geomorphology. We identified a total of 291 beaches across 3000 km of the Pacific coast in Central America and evaluated sand darkness for each beach. The quality of these data was cross-checked with independent information: (i) field observations on some of these beaches (A.M.M.: Guatemala; M.G.: Mexico, Guatemala, and Costa Rica; A.A.C.-G.: Mexico), and (ii) a survey of the literature, e.g., “This dark sand beach is located within the Ostional Wildlife Refuge and measures 3.9 km in length” [28]. We found other evidence indicating that the images carry valuable information: (i) we detected a significant spatial pattern, since two proximate beaches are more similar than randomly expected, and (ii) as expected from geology, we detected a positive relationship of sand lightness with distance to the closest volcano. Yet, a few light sand beaches were observed in proximity to a volcano. Two processes could explain this pattern: (i) the geological signal could have been attenuated when the volcano’s last eruption was ancient, and (ii) flowing and erosion could have transported volcanic material in other directions than the beach.

We found a pivotal temperature of 30.24 °C for temperature-dependent sex determination at constant temperatures (95% credible interval 30.04–30.50 °C) and an upper limit of transitional range of temperatures 5% at 32.16 °C (95% credible interval 31.70–32.68 °C), relatively low values compared to average incubation temperatures recorded in nests in this region, which can exceed 33 °C by a large amount, especially on dark sand beaches [40,41,42]. Hatching success dramatically drops to zero when constant incubation temperatures are over 33.57 °C (Figure 4B). This result is consistent with the observation that hatching success is inversely related to the number of hours spent above 35 °C in olive ridley nests in Playa Coyote, Costa Rica [43]. Similarly, another study reported that the hatching success of *L. olivacea* decreased as incubation temperatures increased above 31 °C in Costa Rica [44].

We found a negative relationship between the proportion of olive ridley nests among the different beaches of Central America and the darkness of the sand (Figure 3). This means that nesting activity was more intense on beaches with lighter sand, higher albedo, and most likely cooler incubation temperatures. According to the natal philopatry hypothesis, the production of females on light-sand, low-temperature beaches would thus be higher overall than on dark-sand, high-temperature beaches. This conclusion is concordant with previous observations for leatherback turtles on Playa Grande beach, Costa Rica, in the same region [45] and experiments conducted on the freshwater turtle *Chrysemys picta* [46]. These studies showed that hotter beaches yielded female-biased primary sex ratios but reduced the total output of female hatchlings. Thus, among our two competing hypotheses, our results support the hypothesis that frequentation of beaches in the region studied is related to differences in hatching success, rather than differences in nest sex ratios resulting from temperature sex determination. This pattern could be a specificity of the East Pacific Central American coast, where feminizing conditions are often associated with lethal incubation temperatures. It is worth mentioning that we hypothesized that no microhabitat selection for temperature-dependent sex determination pattern and lethality had occurred, as demonstrated for green turtles at Ascension Island [47]. However, further work could reveal the same pattern in other nesting areas where thermal limits for viable embryo development might be exceeded (e.g., the Arabian Peninsula). Such studies in other regions and other sea turtle species are warranted to help assess future prospects of actual rookeries in the context of climatic change [48,49].

## 5. Conclusions

After the surprising discovery of the mechanism of temperature-dependent sex determination in turtles [50,51], most predictions about the long-term persistence of sea turtle populations in the face of climate change have focused on the effect of incubation temperature on sex ratios. Other factors involved in population dynamics, such as the actual number of juveniles produced on nesting beaches, have often been overlooked. Our study revealed that when we think about sea turtle population dynamics, we should consider a variety of factors and not only sex ratios.

## Figures and Tables

**Figure 1 animals-11-03168-f001:**
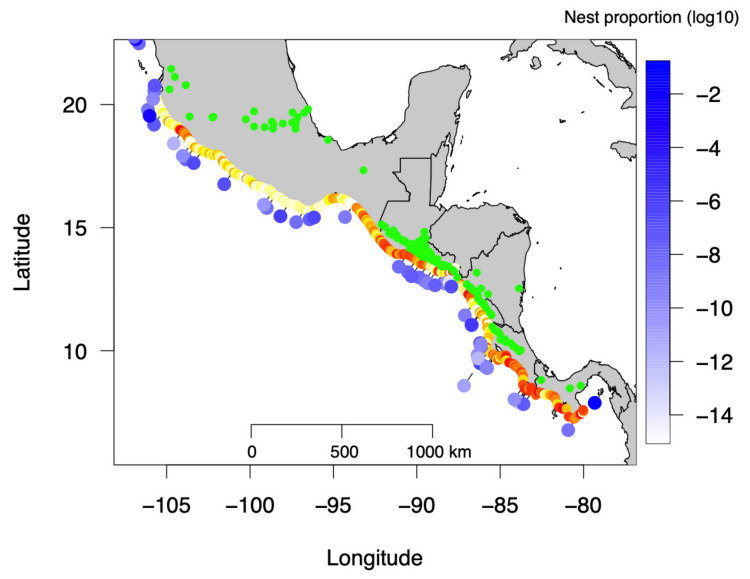
Map of Central America showing the recorded beaches with their estimated darkness (light yellow dots correspond to white sand with a higher albedo and red dots to darker sand with a lower albedo). Green points indicate the position of Holocene volcanos. Blue dots indicate the log10 proportion of olive ridley nests.

**Figure 2 animals-11-03168-f002:**
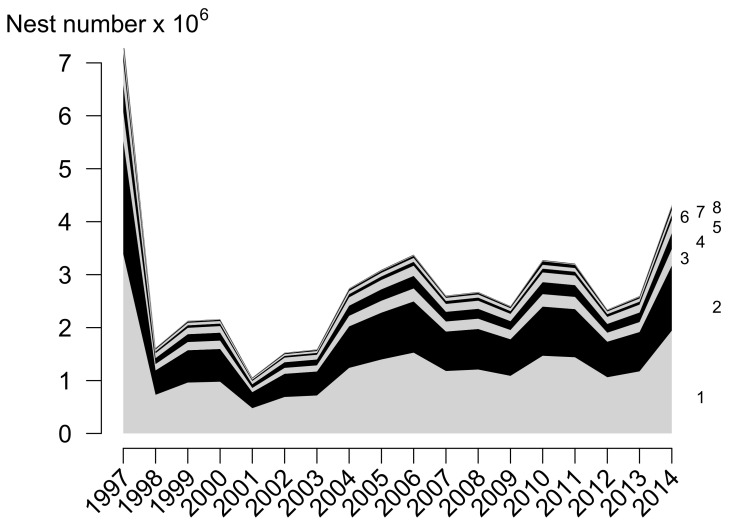
Temporal and spatial distribution of the number of olive ridley nests in Central America. Only eight major beaches are named here from among the 90 beaches used in the present study because the number of nests per year for other ones is too small to be visualized in the figure. Beaches: 1: Santuario Playa de Escobilla; 2: Marinera; 3: Morro Ayuta; 4: La Flor, Carazo; 5: Ixtapilla; 6: RVS Río Escalante-Chacocente; 7: Chacocente; 8: Nancite. They represent 96.7% of the nesting of the analyzed beaches.

**Figure 3 animals-11-03168-f003:**
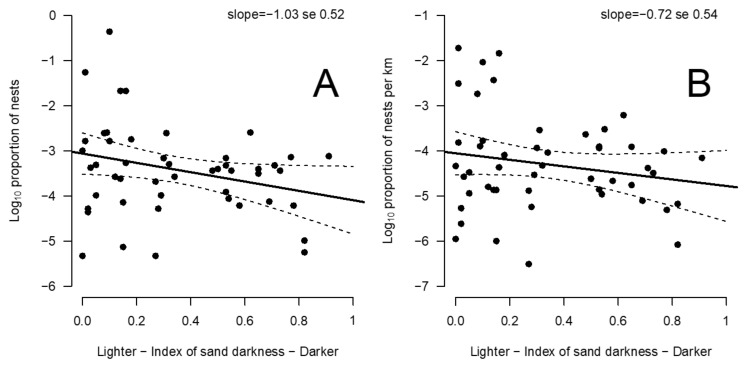
Relationship between sand darkness and (**A**) the proportion of olive ridley nests and (**B**) the proportion of olive ridley nests divided by beach length in Central American beaches. Note that the proportion of nests is log-transformed.

**Figure 4 animals-11-03168-f004:**
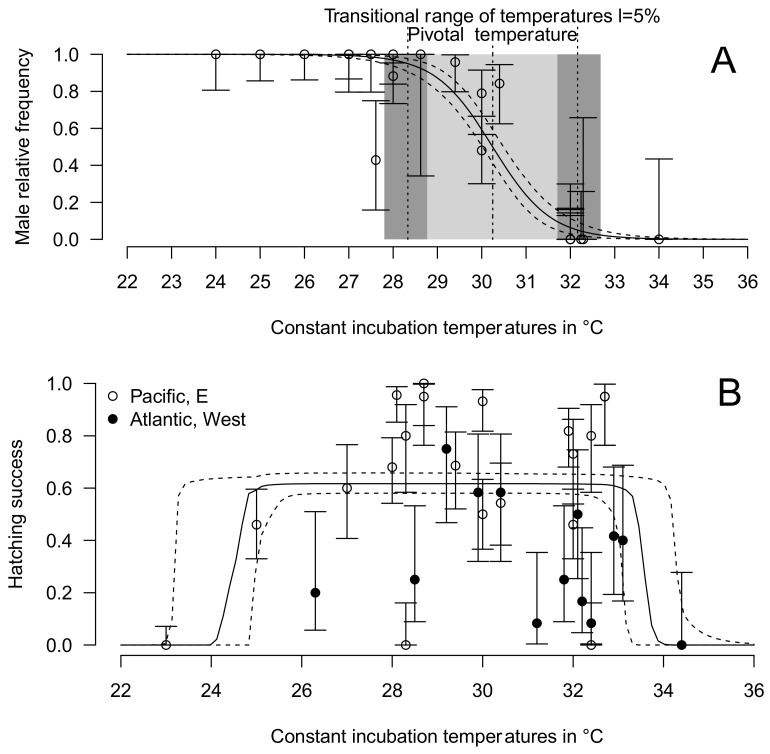
Sex ratio and hatching success at constant incubation temperatures. The thermal reaction norms for (**A**) sex ratio and (**B**) hatching success are shown in solid lines. Light gray temperatures in (**A**) are the range of temperatures that produced a sex ratio from 5% to 95% (transitional range of temperature 5%), and dark gray temperatures are the 95% credible regions for limits of the transitional range of temperature 5%. The temperature that produced 50% of each sex (pivotal temperature) is shown by the interrupted vertical line. In both graphs, the 95% credible regions of the thermal reaction norm are shown with dashed lines. Each point represents a set of eggs from the same origin at a specific constant temperature.

**Table 1 animals-11-03168-t001:** Model selection for temporal distribution of olive ridley nesting activity in Central America.

Temporal Model	AIC	ΔAIC	Akaike Weight
Constant	2254.397	12.53	0.002
Exponential	2247.399	5.53	0.06
Year-specific	2241.867	0	0.94

## Data Availability

Public databases Google Earth Pro V 7.3.2.5776, DatabaseTSD from R package embryogrowth (accessed on 15 October 2019), and the SWOT database (https://www.seaturtlestatus.org, accessed on 20 December 2019) were used in this study.

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
