# Peer review of "Hatching Success Rather Than Temperature-Dependent Sex Determination as the Main Driver of Olive Ridley (Lepidochelys olivacea) Nesting Activity in the Pacific Coast of Central America"

_animals, 2021, doi:10.3390/ani11113168_

Round 1

Reviewer 1 Report

The paper is interesting and adds new information. I suggest the authors to revise the introduction as the first 9 lines of the introduction seem a bit speculative. Also, it is not necessary to specify the meaning of phylopatry.

Author Response

Referee 1

The paper is interesting and adds new information. I suggest the authors to revise the introduction as the first 9 lines of the introduction seem a bit speculative. Also, it is not necessary to specify the meaning of phylopatry.

Re: According to this suggestion, we modified the first paragraph of the introduction

The referee proposes that we don’t specify the meaning of philopatry. We rather think we should keep this definition as it is important to understand the hypothesis of the paper.

Reviewer 2 Report

Page 2 - what is a close beach?

Lines 90-96, what do you tell us all this?  did you break down the beaches to arribada vs solitary beaches?  That would have been interesting to me, and maybe others.  I'm not sure you need all of that text if it isn't relevant to the study.  If relevant then explain in your beach selection model.  You bring up varying densities of nesting on page 3, is this the solitary vs arribada or are you using some other metric to capture nesting density?

I reviewed reference 25, interesting.  This was new to me, so thanks.  Interesting, because we have no way of knowing if development of sex is linked to F1.... but maybe that isn't the point in your study.  Rather you use it as a scale to estimate sex.  

Please try to reword line 235-36.  I read it several times... proximity implies close, so the closer to the volcano the darker the sand... right? So, Albedo should drop?

Figure 3, confused me... why only 4 beaches if you mention 90 in the study.  This lost me, because I felt maybe you were cherry-picking beaches to compare?  Please clarify this figure.

The discussion makes 2 claims that confuse me.  First, Lines 289-294, I'm not sure what this means or why you tell us this?  Second, is lines 304-306.  I fail to understand how a relationship is weak but highly significant?  What is highly significant?  Moreover, if weak yet significant... I believe this needs explaining, because it may indicate that something is wrong with the model... or at least that the model needs a second look.

Author Response

Page 2 - what is a close beach?

Re: We changed the term “close” by “nearby”.

Lines 90-96, what do you tell us all this?  did you break down the beaches to arribada vs solitary beaches?  That would have been interesting to me, and maybe others.  I'm not sure you need all of that text if it isn't relevant to the study.  If relevant then explain in your beach selection model.  You bring up varying densities of nesting on page 3, is this the solitary vs arribada or are you using some other metric to capture nesting density?

Re: We agree that this text is not fully relevant to this study because the model does not split beaches into solitary or arribada beaches. We simplified the text, but we kept the information of high difference in nesting density because it helps to understand figure 2 (in revised version). The new text is (lines 87-92):

L. olivacea employs two nesting strategies depending on aggregation density: 1) solitary nesting like other species, and 2) the group or mass nesting (arribada) behavior in which several thousand females simultaneously nest on the same beach [16]. In Central America, beaches have been monitored for several decades, and density maps of olive ridley crawls, nests, and nesting females are available in both the scientific and gray literature.

I reviewed reference 25, interesting.  This was new to me, so thanks.  Interesting, because we have no way of knowing if development of sex is linked to F1.... but maybe that isn't the point in your study.  Rather you use it as a scale to estimate sex.  

Re: It is indeed used as a scale to estimate sex. No change in the text.

Please try to reword line 235-36.  I read it several times... proximity implies close, so the closer to the volcano the darker the sand... right? So, Albedo should drop?

Re: We added a precision in Material and methods that clarifies this point (lines 149-153):

Two tests were performed to evaluate the accuracy of the color estimation from GoogleEarth pictures. First, it was checked that color of beach sand estimated from GoogleEarth pictures shows a spatial structure. Second, it was tested whether dark beaches are located closer to volcanos than lighter beaches, as expected because basalt material from volcanic origin is darker than material of non-volcanic origin.

Figure 3, confused me... why only 4 beaches if you mention 90 in the study.  This lost me, because I felt maybe you were cherry-picking beaches to compare?  Please clarify this figure.

Re: The 6 beaches (not 4) apparent in figure 3 were the arribada beaches. The other beaches are indeed represented on the graph but cannot be visualized because the number of nests in arribada beaches is much larger, even in log scale. We changed to colors and added this following precision in the legend to figure 3.

Temporal and spatial distribution of the number of olive ridley nests in Central America. Only eight major beaches are named here from among the 90 beaches used in the present study because the number of nests per year for other ones is too small to be visualized on the figure. Beaches: 1: Santuario Playa de Escobilla; 2: Marinera; 3: Morro Ayuta; 4: La Flor, Carazo; 5: Ixtapilla; 6: RVS Río Escalante-Chacocente; 7: Chacocente; 8: Nancite. They represent 96.7% of nesting of the analyzed beaches.

The discussion makes 2 claims that confuse me.  First, Lines 289-294, I'm not sure what this means or why you tell us this? 

Re: This development was transferred from the Discussion to the Material and methods: We hope the new formulation makes it clearer (lines 149-155):

Two tests were performed to evaluate the accuracy of the color estimation from GoogleEarth pictures. First, it was checked that color of beach sand estimated from GoogleEarth pictures shows a spatial structure. Second, it was tested whether dark beaches are located closer to volcanos than lighter beaches, as expected because basalt material from volcanic origin is darker than material of non-volcanic origin. Color estimation was cross-checked with personal observations of the authors for some of these beaches (AMM: Guatemala, MG: Mexico, Guatemala, Costa Rica, AACG: Mexico), and with a survey of the literature.

Second, is lines 304-306.  I fail to understand how a relationship is weak but highly significant?  What is highly significant?  Moreover, if weak yet significant... I believe this needs explaining, because it may indicate that something is wrong with the model... or at least that the model needs a second look.

Re: By “weak relationship”, we designated the low slope of the line on Figure 3. However, this formulation was confusing because the low slope results in part from the logarithmic transformation applied on the number of nests. We hope that the new text no more confuses the reader (lines 251-253):

The relationship between the log10 proportion of nests number and density of nests per km and the sand darkness index was negative (Figure 3A and B): darker beaches tend to have less nesting activity than lighter beaches.

Reviewer 3 Report

This is an important contribution, very well designed and written that deserves to be published in its present form after minor spelling corrections in english.

My congratulations to the authors.

Author Response

This is an important contribution, very well designed and written that deserves to be published in its present form after minor spelling corrections in English.

My congratulations to the authors.

Re: Thanks to the referee. We appreciate

Reviewer 4 Report

The manuscript "Hatching success rather than temperature-dependent sex determination as the main driver of olive ridley (Lepidochelys olivacea) nest density on the Pacific coast of Central America" attempts to relate beach sand color to nest frequency and incubation temperatures, and incubation temperatures to variations in sex ratio and hatchling productivity, therefore ultimately explaining spatial nest frequency patterns using natal philopatry. The logic of their argument in the Introduction is sound - darker beaches should be warmer than their lighter regional counterparts exposed to the same conditions, therefore producing more female hatchlings (assuming the darker sand is not lethally hot) who will return to their natal region to lay the next generation, increasing the expected number of nests at darker beaches. The authors successfully demonstrate the opposite - that lighter beaches, on average, have greater nesting frequencies than darker beaches.

However, I believe there is a significant flaw in the second portion of their manuscript - explaining why the spatial nesting pattern does not conform to expectation. The authors model the reaction norm of the developing embryos to identify which conditions support female-dominant sex ratios and reduced hatchling productivity. But despite these extensive efforts, they do not link this modeling exercise to conditions actually experienced on the study beaches during the period of interest and thus can only further hypothesize as to why the observed spatial nesting pattern does not conform to the expectation. For example, they do not report hatchling productivity at the beaches of interest, so cannot verify that the darker beaches do indeed produce fewer hatchlings, nor do they report observed incubation temperatures in order to determine if differences in hatchling production were related to temperature only as opposed to predation, inundation, etc.

If the manuscript simply reported the use of photogrammetry to document spatial patterns in beach color and the corresponding patterns in nesting frequency, then discussed the implications of this relationship, that would be sufficient. Without a link between the modeling and actual beach conditions, the modeling element should be removed and tailored for its own publication. In addition, the volcano distance-beach color evaluation is unnecessary, poorly supported (i.e., low Pearson correlation coefficient), and does not help answer the core questions of the manuscript, and should be removed.

I have attached an annotated PDF for the authors' review. However, given the significant revisions I believe that would be necessary, I am recommending Rejection at this time.

Author Response

Referee 4

The manuscript "Hatching success rather than temperature-dependent sex determination as the main driver of olive ridley (Lepidochelys olivacea) nest density on the Pacific coast of Central America" attempts to relate beach sand color to nest frequency and incubation temperatures, and incubation temperatures to variations in sex ratio and hatchling productivity, therefore ultimately explaining spatial nest frequency patterns using natal philopatry. The logic of their argument in the Introduction is sound - darker beaches should be warmer than their lighter regional counterparts exposed to the same conditions, therefore producing more female hatchlings (assuming the darker sand is not lethally hot) who will return to their natal region to lay the next generation, increasing the expected number of nests at darker beaches. The authors successfully demonstrate the opposite - that lighter beaches, on average, have greater nesting frequencies than darker beaches.

Re: We agree that it is the main conclusion of our paper.

However, I believe there is a significant flaw in the second portion of their manuscript - explaining why the spatial nesting pattern does not conform to expectation. The authors model the reaction norm of the developing embryos to identify which conditions support female-dominant sex ratios and reduced hatchling productivity. But despite these extensive efforts, they do not link this modeling exercise to conditions actually experienced on the study beaches during the period of interest and thus can only further hypothesize as to why the observed spatial nesting pattern does not conform to the expectation. For example, they do not report hatchling productivity at the beaches of interest, so cannot verify that the darker beaches do indeed produce fewer hatchlings, nor do they report observed incubation temperatures in order to determine if differences in hatchling production were related to temperature only as opposed to predation, inundation, etc.

Re: We agree that we do not produce such data by ourselves, but these relationships are well documented, as exposed below:

  • Relationship between high temperature and sex ratio feminization

Abreu-Grobois, F.A., Morales-Mérida, B.A., Hart, C.E., Guillon, J.-M., Godfrey, M.H., Navarro, E., Girondot, M., 2020. Recent advances on the estimation of the thermal reaction norm for sex ratios. PeerJ 8, e8451.

  • Relationship between sand color and nest temperature

Hays, G.C., Ashworth, J.S., Barnsley, M.J., Broderick, A.C., Emery, D.R., Godley, B.J., Henwood, A., Jones, E.L., 2001. The importance of sand albedo for the thermal conditions on sea turtle nesting beaches. Oikos 93, 87-94.

  • Relationship between high nest temperatures and low hatching success

Valverde, R.A.; Wingard, S.; Gómez, F.; Tordoir, M.T.; Orrego, C.M. Field lethal incubation temperature of olive ridley sea turtle Lepidochelys olivacea embryos at a mass nesting rookery. Endangered Species Research 2010, 12, 77-86, doi:10.3354/esr00296.

Santidrian Tomillo, P., Fonseca, L., Paladino, F.V., Spotila, J.R., Oro, D., 2017. Are thermal barriers "higher" in deep sea turtle nests? PLoS One 12, e0177256.

Binhammer, M.R.; Beange, M.; Arauz, R. Sand temperature, sex ratios, and nest success in olive ridley sea turtles. Marine Turtle Newsletter 2019, 159, 5-9.

Furthermore, we produce the first model to describe the relationship between incubation temperature and hatching success (Figure 5B). It is interesting to note that the lethal temperature is similar to the one measured in the field by Santidrian Tomillo et al. (2017) and Morales-Mérida et al. (2019):

Morales-Mérida, B.A.; Contreras-Mérida, M.R.; Girondot, M. Pipping dynamics in marine turtle Lepidochelys olivacea nests. Trends in Developmental Biology 2019, 12, 23-30.

Accordingly, we made modifications in the text of the introduction and discussion to add references and clarify these points: lines 61-81, lines 303-312.

If the manuscript simply reported the use of photogrammetry to document spatial patterns in beach color and the corresponding patterns in nesting frequency, then discussed the implications of this relationship, that would be sufficient. Without a link between the modeling and actual beach conditions, the modeling element should be removed and tailored for its own publication. In addition, the volcano distance-beach color evaluation is unnecessary, poorly supported (i.e., low Pearson correlation coefficient), and does not help answer the core questions of the manuscript, and should be removed.

The distance to the nearest volcano is only used to confirm that color of GoogleEarth images does provide useful information for quantifying sand albedo. We agree that it is not the core of the paper, so we reduced this part and we removed Figure 2. A low Pearson coefficient is common for Mantel tests.

I have attached an annotated PDF for the authors' review. However, given the significant revisions I believe that would be necessary, I am recommending Rejection at this time._______

Line 4:

Nest density or frequency? You use nest counts (i.e., frequency) in your methods, but switch terminology between frequency and density throughout the manuscript. Since you do not account for beach area in this manuscript, replace all "density" references with "frequency".

We thank the referee for signaling this point. We agree that the term “nest density” was ill-chosen because our index is based on nests counts. Accordingly, we homogenized the text and  replaced “nest density” by “nesting activity” in the title and throughout the manuscript.

We have also included a new figure (Figure 3B) that used the Number of nests per km of beach.

Length of the beaches was calculated using harversine distance between both ends coordinates.

Also, "main" should be capitalized in the title.

Done

Line 36

Why do you have "olive ridley" capitalized throughout the manuscript? Suggest reducing to lowercase.

Done

Line 46

"... whereas others in the same region may have very few nests. The origin of this difference is not well understood, and is sometimes counterintuitive. [1]"

Add additional references to support your argument.

This sentence was removed following recommendation of referee 1.

Line 54

"... beach [3,4], and (iii) female philopatry ..."

Done

Line 55

What about plasticity in nesting beach selection (up to several hundred kilometers)?

Plasticity in nesting beach selection is observed in marine turtles and its spatial scale differs  among species and population. However, this kind of data is not available for Lepidochelys.

Line 74

"... out of the total incident solar radiation ..."

Done

Line 88

Why are the common names for each turtle species capitalized?

Capitalization was removed.

Line 94

Citation for this density-dependent arribada behavior?

Citation added [16]:

Shanker, K.; Abreu-Grobois, A.; Bezy, V.; Briseño, R.; Colman, L.; Girard, A.; Girondot, M.; Jensen, M.; Manoharakrishnan, M.; Rguez-Baron, J.M.; et al. Olive ridleys. The quirky turtles that conquered the world. SWOT 2021, 16.

Line 102

You don't report observed incubation temperatures or hatchling productivity metrics from your study beaches or period, so you can't address Question 2. Your reaction norm modeling is missing connections to conditions experienced on the ground and should realistically be a manuscript in and of itself.

We acknowledge that we don’t have direct measure of temperature over the 90 nesting beaches used in this work. The argument developed to reach our conclusions is based on the literature (see our answer to major point before)

Line 108

Any minimum length requirement for "continuous"?

As long as a stretch of sand was not interrupted by mangrove or vegetation, it was defined as a single beach.

Line 112

Any length or area measurements for each beach recorded? Need this if you are evaluating density rather than frequency. Any covariation in area/size and color?

We estimate density of nest per km and we added this analysis in figure 3B.

 Line 113

So view altitude was variable based on the size of the beach? If so, this means that the ground sampling distance (and related spectral observations) of each pixel varies from beach to beach. Shorter beaches will have smaller ground sampling distances which can better resolve small-scale variations in beach color compared to longer beaches.

We used 85x85=7225 pixels to estimate the modal color of the center of the beach. The use of 7225 pixels permits to minimize a possible effect of beach size.

Line 119 Figure 1

Opposite ends of a color ramp should not have similar colors. It makes them difficult to distinguish on the map.

Color ramp has been changed.

Line 127

Were the constant temperature experimental eggs sourced from light or dark beaches? Any potential for pivotal temperature or TRT adaptation to higher temperatures in darker sand?

We have added this precision in discussion:

It is worth mentioning that we hypothesized that no microhabitat selection for temperature-dependent sex determination pattern and lethality has occurred, as found for green turtles at Ascension Island [47].

Line 130

"The following variables were retrieved from the database: incubation ..."

Done

Line 139

Only one sample area per beach? Were measurements made from most recent image or corresponding to years of nesting data availability? Any assessment of intra-beach variability in space or time?

We always used the most recent beach image as they generally have better resolution. We added this precision (line 106).

Line 142

I assume the lightest and darkest zones are being chosen to represent white and black endpoints for your standardization? If so, please make this clearer. If not, please provide further rationale.

This precision has been added to the text (lines 141-143) :

The lightest and darkest zones of the image that included the entire beach were then selected to represent the color endpoints to standardize color variability across beaches.

Line 154

Why is this volcano-distance analysis necessary? The main purpose of this manuscript is to relate beach darkness to nesting frequency, not explain why some beaches are darker than others. Suggest removing this analysis, particularly since the relationship between color and volcano distance has a weak correlation (Pearson r = 0.13).

The following explanation was added in Material and Methods (lines 150-157):

Two tests were performed to evaluate the accuracy of the color estimation from GoogleEarth pictures. First, it was checked that color of beach sand estimated from GoogleEarth pictures shows a spatial structure. Second, it was tested whether dark beaches are located closer to volcanos than lighter beaches, as expected because basalt material from volcanic origin is darker than material of non-volcanic origin. Color estimation was cross-checked with personal observations of the authors for some of these beaches (AMM: Guatemala, MG: Mexico, Guatemala, Costa Rica, AACG: Mexico), and with a survey of the literature.

A Pearson coefficient as low as 0.13 is common for Mantel tests.

Line 163

Delete "and absent for 1451" as this information is unnecessary.

Done

Line 168:

Is the total number of nests per year available across all years and just the individual beach contributions are missing? Or are some beaches not reported consistently and therefore, the total nest count is incomplete?

We used the SWOT database that reports the annual number of nests per beach. We data were missing for one year, we estimated the missing data using the methodology described in lines 167-204.

Line 201

"... and then a Gaussian distribution was used."

Done

Line 210

Bayesian statistics use a "credible interval". Please correct throughout manuscript.

Change was done throughout the manuscript.

Line 230

A Pearson r of 0.13 is a pretty weak correlation

A low Pearson coefficient is common for Mantel tests.

Line 245

Table 1 is not included

Sorry for this mistake. This table was indeed missing in the submitted manuscript:

Table 1: Model selection for temporal distribution of Olive Ridley nesting activity in Central America.

Temporal model

AIC

∆AIC

Akaike weight

Constant

2254.397

12.53

0.002

Exponential

2247.399

5.53

0.06

Year-specific

2241.867

0.00

0.94

Line 247

"... counts from the SWOT database, ..."

Done

Line 252

Where was beach length recorded, reported, or evaluated in manuscript?

We added this precision (lines 109-110) :

Length of the beaches was calculated using harversine distance between both ends coordinates.

Line 254 – Figure 3

This overlapping text/graphic is difficult to read. Can you left-justify the text on the right edge of the graphic or add a legend in the top right corner?

Graphic colors and legend have been changed.

Line 255

What percent of nesting do these 6 beaches represent?

We added this precision in legend of Figure 3:

They [8 beaches] represent 96.7% of nesting of the analyzed beaches.

Line 263

"East Pacific", not "Pacific East"

Done

Line 273 – Figure 5A

Multiple shades of grey are not visible in 5A.

Corrected.

Line 287

Ground-truthing should be listed in Methods

Done

Line 305

You do not present explicit evidence that nesting beaches during the study period were above thermal tolerances.

We added references to the literature (lines 303-312):

We find a pivotal temperature of 30.24 °C for temperature-dependent sex determination at constant temperatures (95% credible interval 30.04-30.50 °C) and an upper limit of transitional range of temperatures 5% at 32.16 °C (95% credible interval 31.70-32.68 °C), relatively low values compared to average incubation temperatures recorded in nests in this region, which can exceed 33 °C by large, especially on dark sand beaches [40-42]. Hatching success dramatically drops to 0 when constant incubation temperatures are over 33.57 °C (Figure 4B). This result is consistent with the observation that hatching success in inversely related to the number of hours spent above 35°C in olive ridley nests on Playa Coyote, Costa Rica [43]. Similarly, another study reported that hatching success of L. olivacea decreased as incubation temperatures increased above 31°C in Costa Rica [44].

Line 307

"... and an upper limit ..."

Done

Line 308

"range", not "rage"

Done

Line 314

"... olive ridley nests on the Pacific coast of Mexico [30]."

Done

Line 317

Your study does not confirm these previous observations. You do not report hatchling productivity differences observed at darker vs. lighter beaches or how these differences relate to incubation temperatures observed during the study period. Your reaction norm modeling simply hypothesizes that 1) higher temperatures (>33.5 deg C) above the thermal tolerance would theoretically produce only female hatchlings, and 2) that hatchling production is stable (~60%) across the TRT. But this hypothesis is not confirmation.

We agree with this remark. We changed the sentence accordingly (lines 318-320) :

This conclusion is concordant with previous observations for leatherback turtles in Playa Grande beach, Costa Rica, in the same region [45] and experiments conducted on the freshwater turtle Chrysemys picta [46].

Round 2

Reviewer 4 Report

The authors' improvements and responses to criticisms and questions following the initial reviews addressed many of my initial concerns. There are 2 minor corrections (1. In Figure 1, the volcanos are symbolized as circles whereas the caption says triangles, and 2. In Line 306, by large what? the sentence is grammatically incomplete), but otherwise I am recommending Acceptance of the manuscript. Congratulations to the authors.